# How Easily do Irrelevant Inputs Skew the Responses of Large Language Models?

**Siye Wu**♠♣∗ **Jian Xie**♠∗ **Jiangjie Chen**♠ **Tinghui Zhu**♠ **Kai Zhang**♡ **Yanghua Xiao**♠†
♠Shanghai Key Laboratory of Data Science, School of Computer Science, Fudan University
♣Wuhan University ♡The Ohio State University
{siyewu24, jianxie22}@m.fudan.edu.cn, shawyh@fudan.edu.cn

## Abstract

By leveraging the retrieval of information from external knowledge databases, Large Language Models (LLMs) exhibit enhanced capabilities for accomplishing many knowledge-intensive tasks. However, due to the inherent flaws of current retrieval systems, there might exist irrelevant information within those retrieving top-ranked passages. In this work, we present a comprehensive investigation into the robustness of LLMs to different types of irrelevant information under various conditions. We initially introduce a framework to construct high-quality irrelevant information that ranges from semantically unrelated, partially related, and related to questions. Furthermore, our analysis demonstrates that the constructed irrelevant information not only scores highly on similarity metrics, being highly retrieved by existing systems, but also bears semantic connections to the context. Our investigation reveals that current LLMs still face challenges in discriminating highly semantically related information and can be easily distracted by these irrelevant yet misleading content. Besides, we also find that current solutions for handling irrelevant information have limitations in improving the robustness of LLMs to such distractions. All the resources are available on GitHub.

## 1 Introduction

Despite the impressive capabilities of Large Language Models (LLMs) (Brown et al., 2020; Ouyang et al., 2022; Chowdhery et al., 2023) when accomplishing a wide range of tasks, their effectiveness is compromised by inherent limitations rooted in their limited parametric memory, resulting in instances of hallucination or inaccurate responses (Shuster et al., 2021; Ji et al., 2023). Augmented with external retrievers, LLMs demonstrate superior performance by retrieving from external knowledge sources (Lewis et al., 2020; Guu et al., 2020; Borgeaud et al., 2022; Izacard et al., 2023).

However, current retrieval systems are not always reliable since they often provide top-ranked passages indiscriminately that still contain irrelevant information (BehnamGhader et al., 2023; Asai et al., 2024). In real-world Retrieval-Augmented Generation (RAG) applications, retrievers are facing more complex forms of irrelevant information (Cuconasu et al., 2024). Although such irrelevant information scores highly on similarity metrics and may be semantically related to the context, it is irrelevant to answering questions. Even worse, irrelevant information may cause LLMs to change what they have believed, leading to a fabricated answer (Wang et al., 2023). In Figure 1, we give an example to show how such related irrelevant information might distract LLMs, as the misleading information may prompt LLMs to engage in over-reasoning (Hou et al., 2024; Chiang & Lee, 2024).

In this work, we study the robustness of LLMs to irrelevant information. To be specific, we seek to answer the question: ***How well do current LLMs perform when encountering irrelevant information, particularly when it is semantically related?***

---

∗The first two authors contributed equally. Work was done when Siye visited Fudan University.
†Corresponding author.

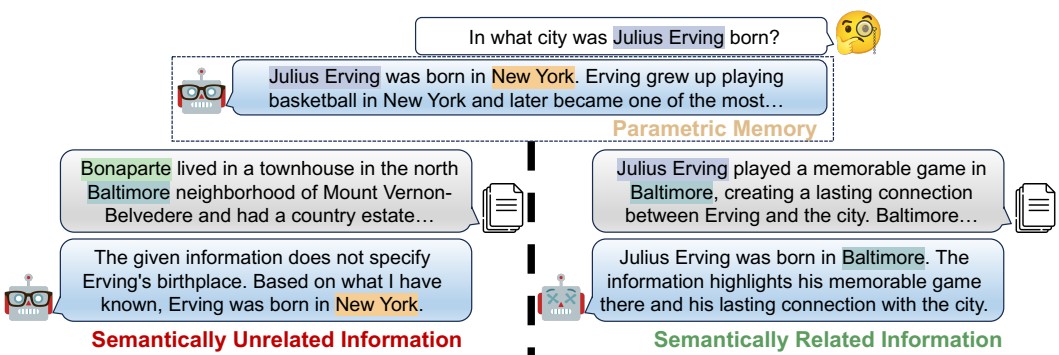

Figure 1: An example of how semantically related irrelevant information distracts LLMs. LLMs are misled by the information due to over-reasoning.

To answer this question, we adopt question answering (QA) tasks for fundamental experiments due to their prevalence in real-world RAG applications (Gao et al., 2023). We first introduce a framework to construct irrelevant information that ranges from semantically unrelated, partially related, and related to questions, and give an analysis that our irrelevant information exhibits high quality, with similarity scores comparable to those of the top-ranked information from Wikipedia, which is easily retrieved by RAG systems. We then systematically assess the robustness of LLMs when faced with irrelevant information, examining their performance under various conditions. We highlight our key findings:

1. Compared to common semantically unrelated irrelevant information, LLMs are more likely to be misled by irrelevant information that is highly semantically related.

2. With the increment of irrelevant information quantity, LLMs are less capable of identifying truly relevant information and are more easily distracted.

3. The robustness of LLMs to irrelevant information varies with the question format, with the free-form format proving to be the most robust.

4. Current strategies intended to improve LLMs' discrimination capabilities result in only marginal, and sometimes even detrimental, enhancements in their ability to accurately identify and disregard irrelevant information.

## 2   Related Work

### 2.1   Retrieval-Augmented Generation

Retrieval-Augmented Generation (RAG) demonstrates impressive abilities in a wide range of knowledge-intensive tasks (Lewis et al., 2020; Guu et al., 2020; Borgeaud et al., 2022; Izacard et al., 2023). LLMs utilize retrieval systems to navigate through external knowledge bases and identify a set of potentially relevant documents, thereby extending beyond the limitations of their parametric memory. Specifically, leveraging dense retriever models (Karpukhin et al., 2020; Gautier et al., 2022) and in-context learning (ICL) (Brown et al., 2020), retrieval-augmented approaches have shown to be remarkably effective in enhancing the capabilities of LLMs (Luan et al., 2021; Mallen et al., 2023; Ram et al., 2023; Shi et al., 2023b). Nonetheless, a challenge persists in the practical deployment of RAG systems, as they indiscriminately surface top-ranked documents that still include irrelevant distractions (BehnamGhader et al., 2023; Wang et al., 2023; Asai et al., 2024; Cuconasu et al., 2024). This issue undermines their utility in real-world applications, where precision and relevance in information retrieval are critical for decision-making processes, such as in medical diagnoses (Zhou et al., 2023). The presence of irrelevant information can lead to inaccurate outcomes, highlighting the need to enhance the reliability of RAG systems.

## 2.2 Robustness to Irrelevant Information

Robustness, which refers to a system's stability when confronted with unexpected inputs (Chang et al., 2023), has been extensively evaluated in previous studies on LLMs (Zhu et al., 2023; Chen et al., 2024). Given its potential to significantly impact model performance, irrelevant information has also attracted attention in the community (Shi et al., 2023a). Prior studies (Shi et al., 2023a; Wu et al., 2024) add specific instruction into prompts, enabling LLMs to better solve math word problems by automatically verifying the irrelevant content within problem descriptions. This approach can be combined with Chain-of-Thought (CoT) prompting method (Wei et al., 2022; Kojima et al., 2022). However, these investigations primarily focus on irrelevant problem descriptions in arithmetic reasoning. In contrast, the challenge of irrelevant information in RAG applications arises more often from retrieved passages. Previous studies often classify low-ranked passages, random passages, and top-ranked passages without ground truth answers as irrelevant information (Yoran et al., 2023; Wang et al., 2023; Yu et al., 2023; Chen et al., 2024). Nonetheless, current advanced RAG systems may effectively filter out such content (Askari et al., 2023). In the real-world scenario, however, semantically related yet irrelevant information, which is highly likely to be retrieved by current systems, remains a challenge. To bridge this gap, our work meticulously constructs high-quality irrelevant information and offers a comprehensive analysis of LLMs' performance across various scenarios. This method enhances our understanding of LLMs' interactions with irrelevant information, thereby providing valuable insights for improving the efficiency and effectiveness of RAG systems.

## 3 Experimental Setup

In this section, we detail the datasets utilized in our work, describe our methodology for constructing high-quality irrelevant information that ranges from semantically *unrelated*, *partially related*, and *related* to questions. Besides, we introduce the metrics used to evaluate the robustness of LLMs to such information.

### 3.1 Datasets

Given the widespread use of question answering (QA) tasks in real-world RAG applications (e.g., New Bing), following previous work (Yoran et al., 2023; Wang et al., 2023; Yu et al., 2023), we employ QA tasks as the foundation for our experiments. Specifically, we focus on entity-centric QA since it is prevalent in RAG scenarios.

- **PopQA** (Mallen et al., 2023): This entity-centric QA dataset comprises 14,000 questions, derived from fact *(subj, relationship, obj)* triples of 16 relationship types in Wikidata. For example, the question, "In what city was Julius Erving born?", is derived from *(Julius Erving, place of birth, New York City)* triples. Furthermore, structured triples facilitate the process of controllable irrelevant information construction, as will be detailed in Section 3.4.

- **EntityQuestions** (Sciavolino et al., 2021): To encompass a wider range of question types in application scenarios, we adopt another widely used entity-centric QA dataset EntityQuestions to broaden the diversity. We exclude relationships that were previously addressed in PopQA to minimize redundancy, yielding 17 distinct relationship types within this dataset. Aligning with the scale of PopQA, we randomly sample 1,500 entries in each relationship for subsequent experiments. Please refer to Appendix A.1 for more details.

### 3.2 Parametric Memory Elicitation

To rigorously evaluate whether LLMs are distracted by irrelevant information, it is essential to first assess their previously internal knowledge free from disturbances. Specifically, following Xie et al. (2023), through closed-book QA format, we extract answers to questions from QA datasets, as well as the corresponding parametric memory from LLMs. For instance, as shown in Table 1, given a question, "In what city was Julius Erving born?",

LLMs are guided to provide a memory answer "New York City" along with background details. This approach ensures that we can observe the robustness of LLMs to irrelevant information by examining if there are any changes in their answers. Furthermore, the elicited parametric memory will serve as one of the pieces of relevant information in the subsequent experiment, leveraging LLMs' inherent confirmation bias to trust their parametric memory (Xie et al., 2023), enhancing the credibility of findings within RAG systems that use LLMs as foundational models.

### 3.3   Information Categorization

In this work, as shown in Figure 2, we categorize information into "Relevant" and "Irrelevant". Specifically, following previous research (Cuconasu et al., 2024), we treat information that leads to or entails an exact answer as relevant, regardless of its correctness. Thus, gold passage and parametric memory passing a series of checks are considered relevant. Unlike relevant information, which suggests an exact answer, irrelevant information cannot support a possible answer and contains entities that distract LLMs. We aim to investigate whether LLMs are robust to these distractions. Previous research has shown that LLMs can be easily distracted by irrelevant information, where even information with no relation to the topics of the questions can mislead them (Shi et al., 2023a). However, there is a lack of detailed analysis concerning the degree of semantic relevance of irrelevant information that affects the performance of LLMs. To address this gap, we introduce a framework for categorizing irrelevant information into three graded levels, aiming to explore its impact in depth. Specifically, we define three distinct levels of irrelevant information: ***Unrelated Information***, ***Partially Related Information***, and ***Related Information***.

Given the vast amount of information stored in databases, retrieving passages with high similarity scores that are nonetheless unrelated to the question topic is inevitable. We categorize such information as ***Unrelated Information***. Furthermore, we aim to do an in-depth analysis of the robustness of LLMs in the wild environment where there is more complex irrelevant information. Such information not only scores highly on similarity metrics but also overlaps with the topics of the ques-

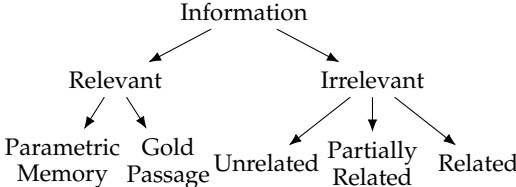

Figure 2: A tree of information categorization in this paper. "Gold Passage" indicates the passage containing gold answers.

tions. We focus on whether LLMs can distinguish between semantically related but misleading information. To answer this question, we incorporate semantically graded related information ***Partially Related Information*** and ***Related Information***.

### 3.4   Irrelevant Information Construction

To construct realistic irrelevant information that is likely to be retrieved in real-world applications, we utilize Wikipedia, one of the largest databases, as our source. Following Mallen et al. (2023), we use a dense retriever, the Contriever model (Gautier et al., 2022), to retrieve the Top 10 passages corresponding to each query. These passages are then used as candidate segments for constructing irrelevant information. Please refer to Appendix A.2 for more processing details. In subsequent parts of this subsection, we use the question "In what city was Julius Erving born?" as an illustrative example to clarify our construction strategy. Accordingly, we present the corresponding *(subj, relationship, obj)* triples of this question as *(Julius Erving, place of birth, New York City)*.

**Unrelated Information**   In Table 1, we give an example of Unrelated Information, which may be retrieved due to their high similarity scores, despite their lack of topical relevance. Specifically, another subject *subj'* "Bonaparte" and its corresponding object *obj'* "Baltimore" are totally unrelated to the question. In order to construct such information, we select a passage from the same relationship (e.g., place of birth) that possesses the highest similarity score, provided it contains another *subj'* and corresponding *obj'*, to serve as the "Unrelated

| **Question:** In what city was Julius Erving born? | | |
|---|---|---|
| **(subj, relationship, obj) Triples: (Julius Erving, place of birth, New York City)** | | |
| **Ground Truth:** New York City | | |
| **A) Parametric Memory:** The city of East Meadow is located in Nassau County on Long Island, New York. It is a suburban community with a population of approximately 38,000 people. Julius Erving, also known as Dr. J, was born there on February 22, 1950. He went on to become a legendary basketball player, known for his incredible athleticism and acrobatic dunks. Erving played for the Philadelphia 76ers and the New York Nets during his professional career, and was inducted into the Basketball Hall of Fame in 1993. | **B) Unrelated Information:** God's mind as shadowed in the workings of the minds of men. Young ladies, if this degree has such meaning for your brothers, what meaning has it for you. Bonaparte lived in a townhouse in the north Baltimore neighborhood of Mount Vernon-Belvedere and had a country estate in suburban Baltimore County, Maryland, which surrounds the city on the west, north and east. His home, Bella Vista, was designed by the architects James Bosley Noel Wyatt, and William G. Nolting, in the prominent... | **C) Partially Related Information:** they married in 2008. Erving has fathered nine children in total. Julius Erving Julius Winfield Erving II (born February 22, 1950), commonly known by the nickname Dr. J, is an American retired basketball player who helped popularize a modern style of play that emphasizes leaping... Baltimore is the most populous city in the U.S. state of Maryland. With a population of 585,708 at the 2020 census, it is the 30th-most populous city in the United States. Baltimore was designated an independent city by... |
| *subj*: Julius Erving, memory answer: New York | *subj'*: C. J. Bonaparte, *obj'*: Baltimore | *subj*: Julius Erving, *obj'*: Baltimore |
| **D) Related Information - Misleading Linkage:** During his illustrious career, Julius Erving played a memorable game in Baltimore, where he dazzled the crowd with his exceptional skills. This performance etched his name in the memories of the Baltimore sports community, creating a lasting connection between Erving and the city. Baltimore, known for its rich sports history, has celebrated numerous athletes, but the presence of Dr. J on their court is a highlight that many basketball enthusiasts in the city still recall fondly. | **E) Related Information - Common Characteristics:** Julius Erving, often known as Dr. J, shared a commonality with C. J. Bonaparte in their dedication to excellence within their respective fields. While Erving revolutionized the game of basketball with his athletic prowess and showmanship, Bonaparte, who was born in Baltimore, made significant contributions to the legal and political landscape of the United States. Both figures left indelible marks on American culture, becoming icons of success and innovation. | **F) Related Information - Fictional Anecdotes:** There's an interesting anecdote that ties Julius Erving to the legacy of C. J. Bonaparte, who was born in Baltimore. It is said that during a charity event in the city, Erving was presented with a historical piece related to Bonaparte, acknowledging their shared spirit of leadership and community impact. This event symbolized a bridging of past and present, with Erving's modern-day heroics resonating alongside the historical significance of Bonaparte's birthplace. |
| *subj*: Julius Erving, *obj'*: Baltimore | *subj*: Julius Erving, *subj'*: C. J. Bonaparte, *obj'*: Baltimore | *subj*: Julius Erving, *subj'*: C. J. Bonaparte, *obj'*: Baltimore |

Table 1: Examples in POPQA. Parametric Memory is elicited from LLMs. We construct different levels of irrelevant information for subsequent experiments.

Information". Existing research on evaluating the robustness of LLMs to irrelevant information has primarily focused on testing their performance at this basic level of irrelevant information (Yoran et al., 2023; Xie et al., 2023).

**Partially Related Information** As shown in Table 1, "Partially Related Information" is structured into two paragraphs. First, from the question corresponding Top 10 passages, we select one that contains *subj* (e.g., Julius Erving) but lacks *obj* (e.g., New York City) as the first paragraph. Such a paragraph provides context about the *subj* yet does not contribute to answering the question. Next, we compute similarity scores between the question and all Wikipedia passages retrieved earlier under the same relationship of questions. We then identify the passage with the highest score that contains the corresponding answer, which is then regarded as *obj'* (e.g., Baltimore), and subsequently acquire a Wiki introduction of *obj'* via Wikipedia-API as the second paragraph. Following these steps, we concatenate these two paragraphs to form "Partially Related Information".

**Related Information** Compared with "Partially Related Information", "Related Information" is highly semantically related to the question yet does not aid in answering it. To develop "Related Information" of high quality, we utilize the triples formed during the "Partially Related Information" stage, introducing additional misleading connections between the subject (*subj*) and the incorrect object (*obj'*) or another subject (*subj'*). Specifically, we create three variants of "Related Information": **1) Misleading Linkage:** This variant focuses on reinforcing the connection between *subj* and *obj'*. In the example in Table 1, Julius Erving and Baltimore are connected through his presence on the court of the city, which enhances the potential for confusion. **2) Common Characteristics:** This variant highlights similarities between *subj* and another *subj'*, where the latter is associated with *obj'*. In the example, common characteristics between Erving and Bonabarte are contributions within their respective fields, thus adding a misleading layer of similarity. **3) Fictional Anecdotes:** This variant creates scenarios involving *subj* and *subj'*, incorporating creative but irrelevant details. In the example, Erving received a historical Bonaparte item in Baltimore, linking their leadership legacies. We utilize GPT-4 Turbo to generate natural language information based on the above-mentioned triples and misleading connections. Please refer to Appendix

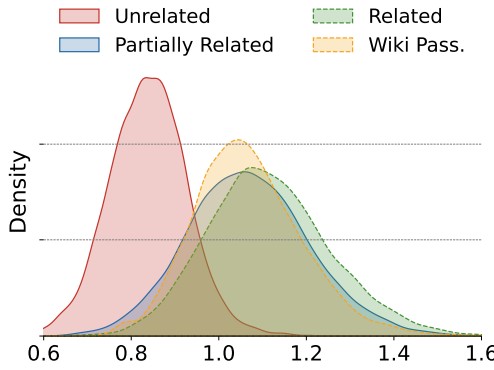

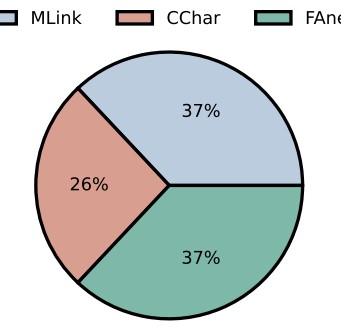

Figure 3: Distribution of similarity scores across different information types for POPQA, with "Wiki Pass." indicating the Top 1 Wikipedia passage specifically retrieved by the Contriever model.

Figure 4: Top-scored variant proportions in **Related Information** for POPQA. "MLink", "CChar", and "FAnec" indicate "Misleading Linkage", "Common Characteristics", and "Fictional Anecdotes", respectively.

A.3 for more details on generating Related Information and Appendix A.4 for more details on the scales of the datasets at each step.

## 3.5 Information Quality Measurement

In order to assess whether the constructed irrelevant information differentiates the semantic relevance, we use the Contriever model to compute the similarity scores between questions and different levels of irrelevant information. We illustrate the results in Figure 3. We find: *1)* Our "Related Information" exhibits similarity scores comparable to those of the top-ranked human-written information from Wikipedia. This distinction is critical as much of previous research has used random or low-ranked passages as irrelevant information for experiments, which might have been filtered by external retrievers. Our approach, by contrast, crafts irrelevant information with higher similarity scores. It is more reasonable in real-world scenarios where RAG systems are more likely to retrieve such information. *2)* The similarity scores for irrelevant information of different levels demonstrate a progressive increase, highlighting the graded nature of our constructed information.

Furthermore, as shown in Figure 4, for "Related Information", the similarity scores across the three variants, each focusing on a unique dimension of misleading content, maintain consistency. These ensure both the diversity and the high quality of our constructed irrelevant information. We provide corresponding measurement figures for ENTITYQUESTIONS in Figure A.1 and Figure A.2 in the Appendix.

## 3.6 Evaluation Metrics

To quantify the impact of irrelevant information interference on LLMs' response changes, we incorporate two specific evaluation metrics:

- **Misrepresentation Ratio.** This metric assesses the rate at which LLMs modify their responses due to the influence of irrelevant information, effectively measuring their propensity to be misled by irrelevant information. [1]

- **Uncertainty Ratio.** This metric calculates how often LLMs indicate uncertainty in their replies (e.g., responses that include phrases like "I'm not sure"). It serves to measure the likelihood of LLMs expressing a lack of confidence in their answers caused by interference of irrelevant information.

---

[1]Whether LLMs rely on parametric memory or gold passages is not considered misrepresentation, as both types of information are deemed "relevant".

| Models | POPQA | | | | | | ENTITYQUESTIONS | | | | | |
| --- | --- | --- | --- | --- | --- | --- | --- | --- | --- | --- | --- | --- |
| | Unrelated | | PartRel. | | Related | | Unrelated | | PartRel. | | Related | |
| | MR | UR | MR | UR | MR | UR | MR | UR | MR | UR | MR | UR |
| GPT-4 Turbo | 8.2 | 9.0 | 8.5 | 15.3 | 15.0 | 9.2 | 4.9 | 15.8 | 4.3 | 12.7 | 10.2 | 10.5 |
| GPT-3.5 Turbo | 5.5 | 72.3 | 10.0 | 59.2 | 22.5 | 28.3 | 3.6 | 66.1 | 5.0 | 37.7 | 11.9 | 26.7 |
| Gemini Pro | 5.3 | 74.2 | 5.9 | 58.8 | 10.3 | 45.3 | 3.3 | 80.7 | 4.1 | 51.0 | 9.5 | 47.8 |
| Llama2-7B | 72.2 | 5.6 | 85.1 | 0.9 | 83.5 | 0.9 | 57.3 | 6.0 | 62.3 | 1.8 | 68.4 | 0.7 |

Table 2: Results of LLMs when confronted with irrelevant information at three different levels of semantic relevance, with PartRel. indicating Partially Related Information, "MR" for misrepresentation ratio, and "UR" for uncertainty ratio, respectively.

To facilitate the answer parsing, following Xie et al. (2023), we adopt the multiple-choice QA format as the primary experimental framework to streamline the process of answer evaluation. To be specific, we offer options including memory answers of LLMs, irrelevant answers generated by human-written templates based on the misleading object *obj'*, and uncertainty options (i.e., "I'm not sure."). Options are shuffled before being presented to LLMs. We provide more details and examples in Appendix A.5. Furthermore, in section 4.3, we investigate the impact of question formats on the performance of LLMs.

## 4 Experiments and Analysis

In this section, we focus on assessing the robustness of LLMs when faced with irrelevant information, examining their performance under various conditions. To be specific, we explore the issue from four distinct perspectives: *1)* **Semantic Relevance** *2)* **Quantity of Information** *3)* **Question Format** *4)* **Limitations of Current Solutions**. We adopt four widely used LLMs for our analysis, including three closed-source LLMs GPT-3.5 Turbo (OpenAI, 2022), GPT-4 Turbo (OpenAI, 2023), and Gemini Pro (G Team et al., 2023), as well as one open-source LLM Llama2-7B (Touvron et al., 2023).

### 4.1 Semantic Relevance

Given the vast amount of information that RAG systems, such as generative search engines, encounter on the Internet, they inevitably face content with varying degrees of semantic relevance. This raises a question: How do LLMs perform when presented with such information? To address this question, we assess the performance of LLMs when confronted with irrelevant information of three different levels of semantic relevance. In our experiments, we introduce a single piece of irrelevant information at a time. For "Related Information", we choose the one with the highest similarity score.

**Highly semantically related information is more likely to distract LLMs.** As shown in Table 2, with the exception of Llama2-7B, LLMs relatively rarely change their memory answers to irrelevant options when facing "Unrelated Information", which is unrelated to the question. However, challenges arise with information that is semantically related to the question. Taking GPT-3.5 Turbo as an example, compared to "Unrelated Information", there is a notable increase in the misrepresentation ratio for "Partially Related Information" and "Related Information", as well as the uncertainty ratio decreases. This indicates that LLMs are more prone to being misled by information that is highly semantically related but irrelevant to the question. Other models also exhibit diminished performance under similar conditions, particularly Llama2-7B. This suggests that current LLMs still struggle with discriminating irrelevant yet highly semantically related information. Additionally, the discussion regarding the uncertainty ratio is detailed in Appendix B.1. From these observations, in order to develop a reliable RAG system, the issue of irrelevant information interference should be taken seriously.

| POPQA | | | | |
|-------|------|------|------|------|
| Models | 1: 0 | 3: 0 | 1: 1 | 3: 1 |
| GPT-3.5 Turbo | 22.5 | 27.4 | 1.7 | 5.5 |
| Llama2-7B | 83.5 | 91.1 | 11.8 | 42.4 |
| ENTITYQUESTIONS | | | | |
| Models | 1: 0 | 3: 0 | 1: 1 | 3: 1 |
| GPT-3.5 Turbo | 11.9 | 11.8 | 0.9 | 1.6 |
| Llama2-7B | 68.4 | 77.9 | 7.2 | 28.4 |

Table 3: Misrepresentation ratio of LLMs under different settings of irrelevant versus relevant information quantities. For example, "3: 1" means three pieces of irrelevant information and one piece of parametric memory.

| POPQA | | | |
|-------|------|---------|------|
| Models | M.C. | Boolean | F.F. |
| GPT-3.5 Turbo | 5.5 | 3.3 | 2.7 |
| Llama2-7B | 42.4 | 20.4 | 14.1 |
| ENTITYQUESTIONS | | | |
| Models | M.C. | Boolean | F.F. |
| GPT-3.5 Turbo | 1.6 | 2.9 | 0.6 |
| Llama2-7B | 28.4 | 14.0 | 4.7 |

Table 4: Misrepresentation ratio of LLMs under different question formats, with "M.C." indicating multiple-choice format, "Boolean" for boolean format, and "F.F." for free-form format. The information ratio is set at 3: 1.

## 4.2 Quantity

In RAG applications, information retrieval often yields a collection of multiple pieces of information, encompassing a mix of both relevant and irrelevant content. In this section, we aim to examine how LLMs perform amidst varying quantities of such information. We adopt corresponding parametric memory as relevant information, which is regarded as convincing by LLMs (Xie et al., 2023). More relevant information settings will be discussed in Section 4.4. Besides, based on Section 4.1, we utilize the remaining distinct variants of "Related Information" to control the quantity of irrelevant information, which is more likely to be retrieved. In order to control the cost, we utilize GPT-3.5 Turbo and Llama2-7B to represent closed-source and open-source LLMs, respectively.

**With the increment of irrelevant information quantity, LLMs are more easily distracted.** As shown in Table 3, when provided solely with irrelevant information, LLMs exhibit a clear trend of increasing tendencies to choose irrelevant answers as the quantity of irrelevant information grows. This suggests that LLMs tend to be distracted by irrelevant but semantically related information, a problem that gets worse with greater quantities of distractions.

**LLMs have a limited capability to identify relevant information.** Furthermore, Table 3 demonstrates a reduction in the misrepresentation ratio upon incorporating relevant information, suggesting that LLMs can indeed discern relevant information. However, even with explicit provision of relevant information, LLMs may still be influenced by the presence of irrelevant information, an issue particularly pronounced in Llama2-7B. This indicates that while LLMs possess the ability to recognize relevant content, their capacity is limited and they remain susceptible to being overwhelmed by irrelevant information—a challenge that could be more acute in the unpredictable conditions of real-world RAG system deployments.

Given the significant quantity of irrelevant information processed by RAG systems in real-world scenarios, our experiments adopt a basic setup where the ratio of irrelevant information to relevant information is 3: 1 unless specified otherwise.

## 4.3 Question Format

Considering various question formations in the real world, in this section, we aim to investigate how question formats influence the performance of LLMs with the interference of irrelevant information. Specifically, in addition to the multiple-choice format, we introduce boolean (true/false) and free-form QA to our experiments. In boolean QA, we ask LLMs to judge the truthfulness of a misleading statement (e.g., "Julius Erving was born in Baltimore"). They are considered distracted if they provide a "true" response. In free-form QA, we present questions to LLMs without providing any options. Due to the difficulty in automatically determining precise answers from LLMs' free-form responses, we utilize

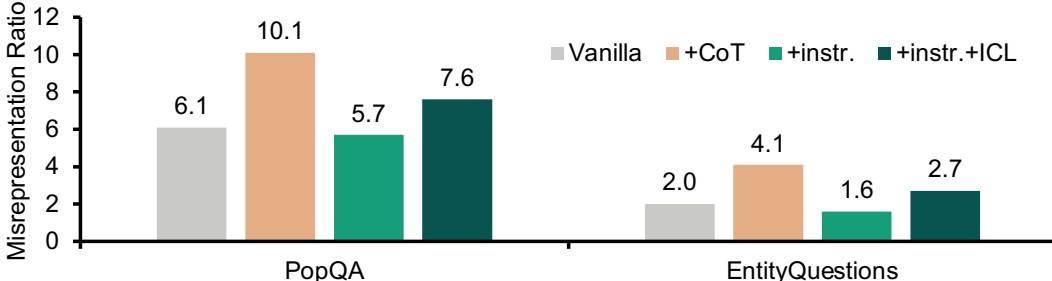

Figure 5: Solution attempts in a more complicated scenario, with all information presented to GPT-3.5 Turbo. Specifically, "+CoT" indicates prompting the model to think step by step, "+instr." adds an instruction "feel free to ignore irrelevant information" into the prompt, and "+ICL" refers to adding examples to guide LLMs in discerning irrelevant information. For a discussion about the model-based fine-tuning method, please refer to Appendix B.5.

GPT-3.5 Turbo to align these responses with specific options. To ensure the accuracy and fairness of GPT-3.5 Turbo's automatic alignment, we conduct human evaluations on 300 randomly selected cases, achieving a 97% accuracy rate. This high level of accuracy validates the fairness and reliability of our assessment method. More details are in Appendix B.2.

**The robustness of LLMs varies with the question formats when faced with irrelevant information.** As shown in Table 4, LLMs are less likely to be misled under free-form and boolean QA compared to multiple-choice format. The presence of an irrelevant answer in the latter format appears to distract LLMs, especially when they are confronted with a vast amount of information that includes misleading content. Such an inconsistent robustness might undermine the truthfulness of RAG systems since the question formats in real-world applications are various. Please refer to Appendix B.3 for an in-depth analysis of the influence of irrelevant answers and case demonstration.

## 4.4 Limitations of Current Solutions

In this section, we aim to investigate whether existing strategies designed to mitigate the impact of irrelevant information are effective in more complex wild environments with multiple pieces of irrelevant information and relevant information. To simulate such a scenario, we present LLMs with a combination of information that includes five pieces of irrelevant content (one "Unrelated Information", one "Partially Related Information", and three "Related Information") alongside two pieces of relevant information (one parametric memory and one gold passage containing the answers). For solutions, we first introduce the CoT method with the prompt "Let's think step by step." because of its proven effectiveness across a range of NLP tasks (Wei et al., 2022; Kojima et al., 2022; Zhu et al., 2024). In addition, following Shi et al. (2023a), we enhance our prompts with an instruction (+*instr.*), "feel free to ignore irrelevant information", aiming to direct LLMs to filter out noise. Furthermore, we assess ICL, by incorporating examples and their labels (relevant/irrelevant) within the prompts, to provide additional guidance for LLMs in navigating through the information. We provide detailed instructions in Appendix B.4.

**CoT might cause over-reasoning due to misleading irrelevant information.** Figure 5 demonstrates that employing the CoT method negatively affects the performance of GPT-3.5 Turbo, particularly implying its tendency to over-reason when presented with misleading information and infer wrong answers due to incorrect reasoning chains. This observation aligns with previous work on math problems, known as over-reasoning (Hou et al., 2024; Chiang & Lee, 2024). For a detailed example of how GPT-3.5 Turbo errs under the influence of irrelevant information, please refer to Appendix B.4. This exploration suggests that while CoT can enhance reasoning depth, its effectiveness varies by model and context, potentially amplifying the risk of distraction by irrelevant details in certain scenarios.

**The "ignoring" instruction has limited effects on LLMs responses.** Previous wisdom (Shi et al., 2023a) has demonstrated adding an instruction that instructs LLMs to ignore irrelevant information could be useful. However, in a more complicated circumstance, we observe a marginal improvement in LLMs' ability against the disturbance of semantically related irrelevant information, which is easy to retrieve due to their high similarity scores.

**Examples given to LLMs may have a negative effect.** We give several examples to guide LLMs in discerning what information is relevant to answering questions and increase their ability to discern distractions. However, as shown in Figure 5, the misrepresentation ratio even gets higher under this setting. This outcome suggests that LLMs struggle to effectively learn from these complex examples and that this form of guidance might, in fact, negatively impact their performance. Please refer to Appendix B.4 for a detailed example analysis.

The findings discussed above highlight that existing strategies fall short of addressing the challenges posed by complex scenarios. This presents a significant obstacle to deploying RAG systems that are both reliable and truthful.

## 5    Conclusion

In this work, we introduce a framework to construct irrelevant information that ranges from semantically unrelated, partially related, and related to questions. The semantically related information exhibits high quality, with similarity scores comparable to human-written information from Wikipedia, which is easily retrieved by RAG systems. Our experiments show that current LLMs still struggle with discriminating highly semantically related irrelevant information under various conditions. And current solutions have limitations in improving the robustness of LLMs to such information. We advocate focused research on mitigating misleading irrelevant interference in the development of reliable RAG systems.

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

# Appendix

# A    Details of Experimental Setup

## A.1    List of Relationship Types and Question Templates

We provide a full list of relationship types and question templates in Table A.1.

| Relationship | Template |
|---|---|
| POPQA | |
| occupation | What is [subj]'s occupation? |
| place of birth | In what city was [subj] born? |
| genre | What genre is [subj]? |
| father | Who is the father of [subj]? |
| country | In what country is [subj]? |
| producer | Who was the producer of [subj]? |
| director | Who was the director of [subj]? |
| capital of | What is [subj] the capital of? |
| screenwriter | Who was the screenwriter for [subj]? |
| composer | Who was the composer of [subj]? |
| color | What color is flag of [subj]? |
| religion | What is the religion of [subj]? |
| sport | What sport does [subj] play? |
| author | Who is the author of [subj]? |
| mother | Who is the mother of [subj]? |
| capital | What is the capital of [subj]? |
| ENTITYQUESTIONS | |
| headquarters location | Where is the headquarter of [subj]? |
| founded by | Who founded [subj]? |
| place of death | Where did [subj] die? |
| performer | Who performed [subj]? |
| location_P131 | Where is [subj] located? |
| location of formation | Where was [subj] founded? |
| record label | What music label is [subj] represented by? |
| country | Which country was [subj] created in? |
| spouse | Who is [subj] married to? |
| creator | Who was [subj] created by? |
| location_P276 | Where is [subj] located? |
| educated at | Where was [subj] educated? |
| notable work | What is [subj] famous for? |
| language | Which language was [subj] written in? |
| child | Who is [subj]'s child? |
| manufacture | Which company is [subj] produced by? |
| owned by | Who owns [subj]? |

Table A.1: Full list of relationship types and question templates in datasets, with [subj] indicating a placeholder for the subject entity.

### A.2 Details of Retrieval Results

We obtained the retrieval results for POPQA using the Contriever model from the repository of the dataset[2]. We also applied the retrieval system to ENTITYQUESTIONS for corresponding results. The corpus we utilized is accessible[3].

### A.3 Related Information Generation

We provide details of the prompts for generating Related Information in Table A.2. To prevent the generation of false context, we add an instruction such as: "However, you are not allowed to claim or hint that 'Julius Erving was/was not born in Baltimore'." In addition, we instruct GPT-4 Turbo to return a "null" response when it encounters difficulties.

---

You are tasked with creating a Question and Answer scenario. Your goal is to compose three distinct paragraphs, each designed to subtly mislead or distract from the actual answer to the question, 'In what city was Julius Erving born?' These paragraphs should be crafted in a way that strongly connects Julius Erving with Baltimore. However, you are not allowed to claim or hint that 'Julius Erving was/was not born in Baltimore'.
Your response is required in JSON format:
{
   "evidence": {
     "paragraph1": "",
     "paragraph2": "",
     ...,
   }
}
The content in the "paragraph" should be "null" if it is hard to compose such a specific paragraph.
1. Find some common characteristics or a connection between Julius Erving and Charles Joseph Bonaparte, mentioning that Charles Joseph Bonaparte was born in Baltimore.
2. Identify a connection between Julius Erving and Baltimore.
3. Create an anecdote involving Julius Erving and Charles Joseph Bonaparte, ensuring that the information "Charles Joseph Bonaparte was born in Baltimore." is mentioned.

---

Table A.2: Prompts for generating Related Information.

### A.4 Scale of Datasets

Scales of two datasets at each step are presented in Table A.3. "Preprocessing" indicates that we exclude data with many possible answers for a clearer assessment.

"Irrelevant Information Construction", as discussed in 3.4, primarily experiences data loss due to the absence of suitable Wikipedia passages for constructing Partially Related Information. GPT-4 Turbo was instructed to return a "null" response when it encounters difficulties in generating three variants of Related Information.

Following Xie et al. (2023), "Parametric Memory Elicitation", as discussed in 3.2, encompasses entailment checking and assessment of answer consistency, which are critical for ascertaining the reliability of LLMs' responses.

### A.5 Creating Options: Process and Illustration

We detail how we create options for the questions and provide an example under multiple-choice QA format in Table A.4. For instance, when assessing GPT-3.5 Turbo for the question "In what city was Julius Erving born?", we utilized its memory answer "Julius Erving was born in East Meadow, New York" from LLMs as the first option. And we use templates,

---

[2]Resources are available at `https://github.com/AlexTMallen/adaptive-retrieval`
[3]`https://huggingface.co/datasets/wiki_dpr`

| POPQA | | | | |
|---|---|---|---|---|
| Initial | | 14,267 | | |
| Preprocessing | | 14,267 | | |
| Irrelevant Information Construction | | 11,234 | | |
| | GPT-4 Turbo | GPT-3.5 Turbo | Gemini Pro | Llama2-7B |
| Parametric Memory Elicitation | 8,457 | 6,487 | 4,687 | 6,246 |
| ENTITYQUESTIONS | | | | |
| Initial | | 25,500 | | |
| Preprocessing | | 24,031 | | |
| Irrelevant Information Construction | | 13,380 | | |
| | GPT-4 Turbo | GPT-3.5 Turbo | Gemini Pro | Llama2-7B |
| Parametric Memory Elicitation | 9,953 | 9,346 | 5,549 | 10,760 |

Table A.3: Scales of two datasets at each step.

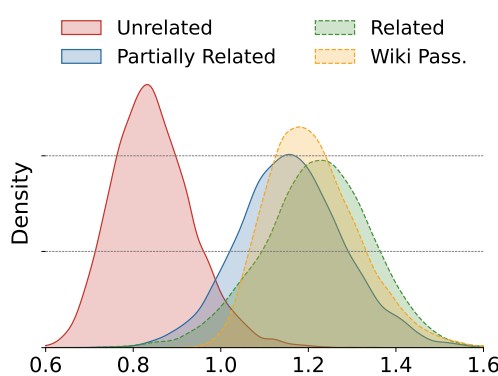

Figure A.1: Distribution of similarity scores across different information types for ENTITYQUESTIONS, with "Wiki Pass." indicating the Top 1 Wikipedia passage specifically retrieved by the Contriever model.

Figure A.2: Top-scored variant proportions in **Related Information** for ENTITYQUESTIONS. MLink", CChar", and FAnec" indicate Misleading Linkage", Common Characteristics", and Fictional Anecdotes", respectively.

similar to A.1, to generate an irrelevant option. Here we use "Baltimore" to generate the second option "Julius Erving was born in Baltimore" based on the relationship "place of birth". Finally, we use the claim "I'm not sure" as the third option. We shuffle the options before presenting them to LLMs.

# B Details of Experiments and Analysis

## B.1 Discussion on Uncertainty Ratio

Table 2 demonstrates that as the semantic relevance of the information increases, the uncertainty ratio decreases. To explore this phenomenon in detail, we sample 100 data points from ENTITYQUESTIONS to observe GPT-3.5 Turbo's responses. The results are shown in Table B.5. We find that LLMs might choose the uncertainty option for security reasons, especially when facing Unrelated Information.

Specifically, in uncertain situations when facing Unrelated Information, 57 out of 73 responses explain "The given information doesn't provide any details about the question", while the rest simply state the option. This indicates that GPT-3.5 Turbo identifies Unrelated Information but expresses uncertainty for conservative reasons.

According to the given information and your knowledge, choose the best choice from the following options.

Information:

1. There's an interesting anecdote that ties Julius Erving to the legacy of Charles Joseph Bonaparte, who was born in Baltimore. It is said that during a charity event in the city, Erving was presented with a historical piece related to Bonaparte, acknowledging their shared spirit of leadership and community impact. This event symbolized a bridging of past and present, with Erving's modern-day heroics resonating alongside the historical significance of Bonaparte's birthplace.

Question:

In what city was Julius Erving born?

Options:

A. I'm not sure.

B. Julius Erving was born in Baltimore.

C. Julius Erving was born in East Meadow, New York.

Answer:

Table A.4: An example under multiple-choice QA format when assessing GPT-3.5 Turbo.

|  | Unrelated | PartRel. | Related |
|---|---|---|---|
| Uncertainty Count | 73 | 39 | 22 |

Table B.5: GPT-3.5 Turbo chooses uncertainty options when facing different information.

### B.2 Manual Evaluation of GPT-3.5 Turbo Accuracy of Free-form Question Alignment

To assess GPT-3.5 Turbo's automatic alignment accuracy, three authors independently evaluated the same 300 randomly selected examples. The process involved identifying the claim that best matched the given options, with the majority consensus among authors defining the ground truth. The comparison of the automatic alignment's outcomes with consensus-ground truths revealed an accuracy rate of 97%, showcasing the high precision.

### B.3 Question Format Influence on Responses of LLMs

We demonstrate the variability in responses of LLMs (e.g., GPT-3.5 Turbo) to various question formats, as shown in Table B.8. Irrelevant information and parametric memory are marked with different colors. We shuffled options and information before experiments.

In the example, when asked "Who was the screenwriter for The Man?" in a multiple-choice QA format, GPT-3.5 Turbo chooses an irrelevant answer (i.e., "C. Gore Vidal is the screenwriter for The Man"). Without the multiple-choice options, GPT-3.5 Turbo relies on its parametric memory (i.e., "The screenwriter for The Man was Jim Piddock"). In contrast, when the question is formatted as a boolean QA, the model exhibits uncertainty in response.

### B.4 Details of Limitation of Current Solution.

We illustrate how CoT tends to over-reason with irrelevant information, and how ICL similarly selects the irrelevant answer. Exemplars in the ICL prompting method are provided in Table B.7 and details are provided in Table B.9 and Table B.10. In this example, all five pieces of irrelevant information are associated with one answer "Carl Gustaf Wrangel". Options and information are shuffled before being presented to LLMs.

In a standard scenario (i.e., Vanilla), GPT-3.5 Turbo relies on its parametric memory, identifying Åkerö Castle was founded by the Swedish nobleman and statesman, Axel Oxenstierna. In contrast, when prompted to think step by step, and based on its inference from the detail that Carl Gustaf Wrangel was a prominent figure in Swedish noble history and had extensive involvement in the development of numerous estates and castles throughout

Sweden, GPT-3.5 Turbo is misled into selecting the irrelevant answer (i.e., Åkerö Castle was founded by Carl Gustaf Wrangel). In addition, despite being instructed to ignore irrelevant information and provided with exemplars, GPT-3.5 Turbo fails to differentiate the irrelevant details, ultimately selecting the irrelevant answer.

## B.5  Results of Fine-tuning Method

In our paper, we focus on prompted-based methods to help discern irrelevant information. As fine-tuning is a proven method to enhance the robustness of LLMs to irrelevant information (Yoran et al., 2023), we additionally sample 100 data points to test how llama2-13b and llama2-13b-peft-nq-retrobust (+ft.) perform with constructed related information. As shown in Table B.6, the misrepresentation ratio (MR) indicates that the fine-tuned model shows improved robustness, but remains high, suggesting challenges still exist.

|                  | MR |
|------------------|----|
| llama2-13b       | 61 |
| llama2-13b + ft. | 25 |

Table B.6: Performance of Llama2-13b and fine-tuned models on 100 sampled data points.

---

{EXEMPLARS}
Example:
Question:
In what city was Alfred Jefferis Turner born?
Information:
1. The city of London has a rich history and is known for its cultural landmarks such as the Tower of London, Buckingham Palace, and the British Museum. It is also a hub for finance...
Label: Relevant
2. The connection between Alfred Jefferis Turner and Great Yarmouth can be traced through the intricate web of Victorian-era scientific communities. During this period...
Label: Irrelevant
3. An anecdote often shared among historians of science recounts a meeting between Alfred Jefferis Turner and Dawson Turner. The two men, both passionate about the...
Label: Irrelevant
4. Alfred Jefferis Turner, a notable figure in the field of entomology, shared more than just a surname with the esteemed banker and botanist Dawson Turner...
Label: Irrelevant
5. Alfred Jefferis Turner–located in the heritage building Fairy Knoll in Ipswich. Doctor Jefferis Turner studied medicine at University College, London graduating with first...
Label: Irrelevant
6. Edward Drinker Cope–museums, zoos, and gardens. Cope's interest in animals became apparent at a young age, as did his natural artistic ability. Alfred intended to...
Label: Irrelevant
7. Alfred Jefferis Turner Alfred Jefferis Turner (3 October 1861 Canton – 29 December 1947 Brisbane, Australia) was a pediatrician and noted amateur entomologist. He was...
Label: Relevant

---

Table B.7: Exemplars in ICL prompting method.

| Question: Who was the screenwriter for The Man? |
| --- |
| Memory Answer: The screenwriter for The Man was Jim Piddock. |
| Irrelevant Answer: Gore Vidal is the screenwriter for The Man. |

{INFORMATION}:

1. Gore Vidal, an esteemed novelist and playwright, had a profound impact on the political dramas of his time. His work often explored the nature of political ambition and the American political system. While 'The Man' shares these preoccupations and stands as a notable example of the genre, it is the sharp dialogue and the exploration of ethical dilemmas in 'The Best Man' that truly bears the hallmark of Vidal's writing style.

2. There's an interesting anecdote that highlights the cultural impact of political dramas during the 1960s. During a screening of 'The Man,' a viewer was overheard remarking on the striking resemblance of the film's narrative to that of 'The Best Man.' This prompted a discussion among the audience about the latter's screenwriter, Gore Vidal, and his talent for capturing the essence of political intrigue. The conversation soon turned into a debate about the role of screenwriters in shaping the political discourse of the time.

3. Jim Piddock is a British actor, writer, and producer who has worked on various films and television shows. He has written and produced several comedy films, including The Man, which was released in 2005. Piddock has also acted in numerous films and TV shows, such as Best in Show, A Mighty Wind, and The Five-Year Engagement. His experience in the entertainment industry has made him a well-respected figure in the world of comedy and screenwriting.

4. When examining the political dramas of the 1960s, one cannot help but notice the thematic similarities between 'The Man' and another significant film of the era, 'The Best Man.' Both movies delve into the intricacies of political power and the moral compromises often made by those in the highest echelons of government. 'The Best Man' is particularly renowned for its sharp and incisive script, which was penned by the acclaimed writer Gore Vidal, known for his keen insight into the political landscape of the United States.

Multiple-Choice:
According to the given information and your knowledge, choose the best choice from the following options.
Information:
{INFORMATION}
Question:
Who was the screenwriter for The Man?
Options:
A. The screenwriter for The Man was Jim Piddock.
B. I'm not sure.
C. Gore Vidal is the screenwriter for The Man.
Answer:
Response of GPT-3.5 Turbo: C. Gore Vidal is the screenwriter for The Man.

Boolean:
According to the given information and your knowledge, determine whether the statement is true or false.
Information:
{INFORMATION}
Statement:
Gore Vidal is the screenwriter for The Man.
Is the statement true or false?
Response of GPT-3.5 Turbo: There is not enough information to determine the veracity of the statement. While the information provided mentions Gore Vidal's involvement in 'The Best Man,' it does not explicitly state whether he was the screenwriter for 'The Man'.

Free-Form:
According to the given information and your knowledge, answer the question.
Information:
{INFORMATION}
Question:
Who was the screenwriter for The Man?
Answer:
Response of GPT-3.5 Turbo: Jim Piddock was the screenwriter for The Man.

Table B.8: An example of how question formats influence responses of GPT-3.5 Turbo.

| |
|---|
| Question: Who founded Åkerö Castle? |
| Ground Truth Answer: Åkerö Castle was founded by Carl Gustaf Tessin. |
| Memory Answer: Åkerö Castle was founded by the Swedish nobleman and statesman, Axel Oxenstierna. |
| Irrelevant Answer (1): Åkerö Castle was founded by Carl Gustaf Wrangel. |

{INFORMATION}:
1. The connection between Åkerö Castle and Carl Gustaf Wrangel is one that is woven through the fabric of Swedish noble history. Wrangel, a prominent figure of his time, was known for his extensive involvement in the development of numerous estates and castles throughout Sweden. His influence was so widespread that it touched upon many aspects of noble life, including the arts, military, and architecture, much like the influences seen in the construction and design of Åkerö Castle.
2. Axel Oxenstierna was a prominent figure in Swedish politics during the 17th century. He served as Chancellor of Sweden from 1612 until his death in 1654. In addition to his political career, Oxenstierna was also a wealthy landowner and built several castles and manors throughout Sweden, including Åkerö Castle. The castle was completed in 1637 and served as a summer residence for Oxenstierna and his family. Today, Åkerö Castle is a popular tourist attraction and is open to the public for guided tours.
3. Gripenberg Castle–commemorate her. By the end of the 17th century the castle was bought by Samuel von Söderling and today remains in the possession of his family. Gripenberg Castle Gripenberg Castle (Swedish: "Gripenbergs slott") is a wooden manor house near Tranås in Småland, Sweden. It is considered to be the biggest wooden castle in Sweden and one of the oldest that remain today as well. Gripenberg Castle was built in 1663 as a hunting seat for the field marshal Carl Gustaf Wrangel. Its architect is unknown, but there is some reason to believe that it might have been Nicodemus Tessin the
4. it belonged to Anna Bielke. In 1660, about a hundred years later, the building was damaged by fire and never completely restored. In 1748 Carl Gustaf Tessin bought the estate. He ordered the old, damaged buildings to be demolished and commissioned a new building to be built to the designs by Carl Hårleman (it was built in 1752-1757). Louis-Joseph Le Lorrain was commissioned to design the interiors. Åkerö Manor Åkerö Manor is a manor in Södermanland, Sweden. Although an estate with a history going back to the Middle Ages, the presently visible manor house complex was commissioned in 1748 (completed
5. Åkerö–Åkerö 'Åkerö', 'Akero', or 'Okera', is an old apple cultivar of presumed Swedish origin, but possibly introduced from the Netherlands. It is a dessert apple with an aromatic flavor. 'Akero' was first described by pomologist Olof Eneroth in 1858, parentage is unknown. It is probably named after the Åkerö Manor located south of Stockholm, Sweden where according to some it was found originally as a seedling. Today it is still cultivated in Sweden and is regarded as the best Swedish apple. Was once very popular in Scandinavia and Northern Germany. The tree is hardy and vigorous with an upright growing Fältmarskalk Carl Gustaf Wrangel (also Carl Gustav von Wrangel; 23 December 1613 – 5 July 1676) was a Swedish statesman and military commander who commanded the Swedish forces in the Thirty Years' War, as well as the Torstenson, Bremen, Second Northern and Scanian Wars. A Baltic German, he held the ranks of a Field Marshal, Commander-in-Chief of the Swedish forces in Germany (1646–1648), and Lord High Admiral of Sweden (from 1657). Wrangel was Governor-General of Swedish Pomerania (1648–1652 and 1656–1676) and, from 1664, Lord High Constable of Sweden and a member of the Privy Council. He held the title of
6. There is an intriguing anecdote that captures the essence of the Swedish nobility's passion for grand estates during the 17th century. It is said that the owners of Åkerö Castle and Gripenberg Castle, the latter founded by Carl Gustaf Wrangel, would hold a friendly rivalry to host the most lavish banquets and balls. Guests would often compare the two, noting the similarities in their architectural beauty and the sophistication of the events, a testament to the era's opulent lifestyle.
7. Åkerö Castle, with its majestic architecture and historical significance, shares a remarkable resemblance to the grandeur of Gripenberg Castle. Both castles are exemplary representations of the Swedish architectural prowess during the 17th century, a period when the nobility invested heavily in constructing lavish residences. It is during this era that Gripenberg Castle was founded by the influential Carl Gustaf Wrangel, a man known for his taste in opulence and his contributions to the Swedish landscape of stately homes.

Table B.9: (1) An example explaining the limitation of current solutions.

{OPTIONS}:
A. Åkerö Castle was founded by Carl Gustaf Wrangel.
B. I'm not sure.
C. Åkerö Castle was founded by the Swedish nobleman and statesman, Axel Oxenstierna.
D. Åkerö Castle was founded by Carl Gustaf Tessin.

---

Vanilla:
According to the given information and your knowledge, choose the best choice from the
following options.
Information:
{INFORMATION}
Question:
Who founded Åkerö Castle?
Options:
{OPTIONS}
Answer:

Response of GPT-3.5 Turbo:  Åkerö Castle was founded by the Swedish nobleman
and statesman, Axel Oxenstierna.

---

Vanilla +CoT:
According to the given information and your knowledge, choose the best choice from the
following options.
Information:
{INFORMATION}
Question:
Who founded Åkerö Castle?
Options:
{OPTIONS}
Answer:
Let's think step by step.

Response of GPT-3.5 Turbo:  First, we need to gather information related to the
founding of Åkerö Castle. From the given information, we can see that option A states that
Åkerö Castle was founded by Carl Gustaf Wrangel, who was a prominent figure in Swedish
noble history and had extensive involvement in the development of numerous estates
and castles throughout Sweden. On the other hand, option C mentions Axel Oxenstierna,
who was a wealthy landowner and built several castles and manors throughout Sweden,
including Åkerö Castle, but it does not state that he founded it. Option D talks about Carl
Gustaf Tessin, who bought the estate much later and commissioned a new building to be
built. Option B is not a valid answer as it does not provide any information. Therefore, the
correct answer is A. Åkerö Castle was founded by Carl Gustaf Wrangel.

---

Vanilla +instr. +ICL:
According to the given information and your knowledge, choose the best choice from the
following options. Feel free to ignore irrelevant information.
{EXEMPLARS}
Information:
{INFORMATION}
Question:
Who founded Åkerö Castle?
Options:
{OPTIONS}
Answer:

Response of GPT-3.5 Turbo: A. Åkerö Castle was founded by Carl Gustaf Wrangel.

---

Table B.10: (2) An example explaining the limitation of current solutions.

