# OpenReview forum: "How Easily do Irrelevant Inputs Skew the Responses of Large Language Models?"
_colmweb.org/COLM/2024/Conference — COLM_

### Official Review · Reviewer_jd2d · 2024-05-09

**Rating:** 5
**Confidence:** 4
**Ethics Flag:** 1

**Summary:**

This paper investigates the robustness of LLMs to irrelevant information, particularly when such information is semantically related but irrelevant to the query. The study involves constructing various types of irrelevant information, ranging from semantically unrelated to related, and assessing their impact on the performance of LLMs.

**Questions To Authors:**

see above

**Reasons To Accept:**

1. The robustness of LLMs to irrelevant information is a very critical issue in the field.
2. The authors provide a fine-grained analysis of the impact of irrelevant information based on its semantic relatedness to the query.
3. The structure of the article is well organized.

**Reasons To Reject:**

- The most important question is: the definition of "irrelevant information" in this paper seems to be unclear.
	- It is not well-explained why "semantically related information" can be considered "irrelevant". The distinction between semantically related information and concepts like incorrect contexts, misinformation, or conflicting knowledge, as mentioned in recent works [1,2,3,4], is not adequately addressed.
	- Why is the "Parametric Memory" in Figure 2 considered "relevant"? After all, Parametric Memory might not necessarily contain the correct answer.

- In Table 4, the impact of question formats on the Misrepresentation ratio is quite apparent, especially for Llama2-7B. The free-form format clearly aligns more closely with real-world application scenarios. Why then does this paper primarily use the M.C. format as described in Section 3.6?
	- In Sections 4.1 and 4.4, the authors do not seem to mention the settings for question formats, and my understanding is that they default to using the M.C. format as discussed in Section 3.6. Perhaps the authors should also clearly specify the question formats used in each section. This has caused me a bit of confusion.
- What should an ideal LLM behave? Should it adhere strictly to its own knowledge, completely ignoring external input? Or should it accurately identify the correct piece of knowledge from multiple external sources?
	- A more specific question about this, in section 4.2, "We adopt corresponding parametric memory as relevant information"? This is hypothesis unreasonable as parametric memory might be incorrect. It may be more reasonable to use the golden passage as relevant information.
	- The authors' use of parametric memory as relevant information seems to suggest that the model should always trust its own knowledge. If that is the case, why do we still need RAG?
- The details or a formal definition of the Misrepresentation Ratio need to be provided more thoroughly. This is crucial for the reliability of the conclusions in the paper. For example, in section 4.4, if the LLM disregards its own parameter knowledge and relies on the golden passage, is this included in the Misrepresentation Ratio?
-  Some Questions Regarding Section 4.4

	- Section 4.4 considers scenarios where multiple types of irrelevant content coexist. The authors only published the results of the Misrepresentation Ratio. Perhaps what is of greater interest is: which specific passage LLMs choose to trust, or which specific irrelevant content misleads LLMs (“Unrelated Information”, “Partially Related Information”, or “Related Information”).
	- Does Section 4.4 ensure that all five types of contexts provide different knowledge? If any two types of knowledge are the same, it could likely affect the model's behavior.
	- The conclusion that "CoT might cause over-reasoning due to misleading irrelevant information" is quite interesting. Has this conclusion been mentioned in other articles that you cited, such as [5]? Is this finding completely novel?
	- Additionally, sec 4.4, "To simulate such a scenario, we present LLMs with a combination of information that includes five pieces of irrelevant content (one 'Unrelated Information', one 'Partially Related Information', and **three 'Related Information'**)." Why including more 'Related Information' than 'Unrelated Information' and 'Partially Related Information'?
	- I suggest that the authors explain the concept of "over-reasoning" and why it is an undesirable situation. This concept is not very familiar to me at least.


[1] Pan Y, Pan L, Chen W, et al. On the risk of misinformation pollution with large language models.	EMNLP 2023

[2] Tan H, Sun F, Yang W, et al. Blinded by Generated Contexts: How Language Models Merge Generated and Retrieved Contexts for Open-Domain QA?. arXiv 2024.

[3] Xu R, Qi Z, Wang C, et al. Knowledge Conflicts for LLMs: A Survey. arXiv 2024.

[4] Adaptive chameleon or stubborn sloth: Revealing the behavior of large language models in knowledge conflicts. ICLR 2024

[5] Over-reasoning and redundant calculation of large language models. EACL 2024

---

> ### Author Rebuttal · Authors · 2024-05-31
>
> Thanks for your efforts in reviewing!
> > Definition
>
> We treat information leading to (entailing) an exact answer as relevant, regardless of its correctness. Thus, gold passage, parametric memory passing double-check[1],  and conflicting knowledge (not used in our paper) are considered relevant.
>
> Please see the discussion about "irrelevant information" in the 1st response to Reviewer rBEp due to word limits.
>
> > M.C. format
>
> Please see the 3rd response to Reviewer rBEp. As noted in Sec3.6, all experiments are under M.C. format except those in Sec4.3. Will state it clearly further.
> > LLMs behavior
>
> Our focus is on investigating irrelevant information. Since irrelevant information doesn't help answer questions and may cause distractions, LLMs should be robust to them.
> - The reason we regard parametric memory as relevant is explained above. As LLMs tend to trust their parametric memory[1], poor performance with parametric memory as relevant information may indicate worse performance with gold passage.
> - We don't suggest that models should always trust their own knowledge. For retrieved information, LLMs should discern the irrelevant content and focus on the truly relevant one.
>
> > MR
>
> MR assesses the rate at which LLMs select irrelevant options influenced by irrelevant information. In Sec4.4, whether LLMs rely on the parametric memory or gold passage is not included in MR as they are both relevant.
> > Questions about 4.4
>
> - 1:  Good point! We sample 100 data points, reconstructing corresponding irrelevant information to ensure different information contains different options.  GPT3.5 tends to trust its parametric memory and its robustness decreases with increasing semantic relevance.
> |Mem.Ans.|Truth.|Unrel.Opt.|PartRel.Opt.|Rel.Opt.|Uncert.|
> |-|-|-|-|-|-|
> |63%|16%|1%|2%|5%|13%|
> - 2, 4: "PartRel. Info" and "Related Info" contain the same irrelevant entities, but this shouldn't affect LLMs’ behavior as they don't support specific answers. Their contexts differ, with only the irrelevant entities overlapping. "Related Info" has higher similarity scores to be retrieved, so we provide more of it.
> - 3, 5: LLMs may do unnecessary reasoning on unknown variables, making errors in math problems[2]. We first find this issue in RAG with irrelevant information: LLMs may infer wrong answers due to incorrect reasoning chains on irrelevant information, which is undesirable. Will state it clearly further.
>
> [1] https://arxiv.org/abs/2305.13300
>
> [2] https://arxiv.org/abs/2401.11467

---

> > ### Comment · Reviewer_jd2d · 2024-06-05
> >
> > I thank the authors for their response and decide to keep my score.

---

> > > ### Author Response · Authors · 2024-06-06
> > >
> > > Thank you for reading our response. We appreciate your insightful feedback and suggestions, as they help us improve the quality and clarity of our paper. We will include the discussion content in the next revision. If you have any other questions, please feel free to follow up at any time.

---

### Official Review · Reviewer_FXdc · 2024-05-09

**Rating:** 7
**Confidence:** 3
**Ethics Flag:** 1

**Summary:**

Language models are trained to use relevant information in query responses. This paper focuses particularly on the the ability of RAG LLMs to discriminate relevant information from other semantically related information, and on particular strategies used to improve performance on RAG tasks, by deliberately constructing semantically related distractors and evaluating on those. One model in particular, GPT-3.5 Turbo, was then tested to to see whether performance is improved by chain-of-thought (CoT) strategies, additional instruction, or additional "in context" labeled examples (ICL).

Results indicate, as expected, that LLMs relatively rarely change their memory answers to irrelevant options when facing “Unrelated Information”, but that they are often distracted by partially related information. Methods commonly used to address this problem, surprisingly, can have a negative impact. With one of the best performers, GPT-3.5, results in Figure 5 indicate that CoT strategies have a negative impact, and even ICL can sometimes have a negative impact.

The negative impact of CoT is perhaps the most surprising result, and the authors attribute it to "over-reasoning". An example supporting this conjecture is analyzed in appendix B3.

For future work, it would be valuable to have a benchmark dataset to assess progress on these issues, and to allow a more comprehensive item analysis than can be provided by a examples in the appendix.

This research is well-situated with respect to the literature and clearly presented.

**Questions To Authors:**

To facilitate replication of these results, it would be valuable to make components of the data construction and analysis available.

**Reasons To Accept:**

Deliberately constructed distractors is an interesting approach to assessing model robustness, and the negative impact of CoT and ICL is surprising -- further study should elucidate what is happening in cases like these.

**Reasons To Reject:**

While the work is fairly clearly explained, it would be a very considerable task to replicate.

---

> ### Author Rebuttal · Authors · 2024-05-31
>
> Sincerely thanks for your comments! We also feel surprised about the negative impact of CoT and ICL, and hoping these findings can enlighten the following study.
>
>
> > To facilitate replication of these results, it would be valuable to make components of the data construction and analysis available.
>
> We have posted the prompts used in this paper in the Appendix, and we will release our code and data to facilitate replication of our results upon acceptance.

---

> > ### Comment · Reviewer_FXdc · 2024-06-04
> >
> > Thanks for your response. It would be difficult to replicate these results without the code and data.

---

> > > ### Author Response · Authors · 2024-06-06
> > >
> > > Sincerely thank you for your engagement during the discussion. We will release all the resources upon acceptance.

---

### Official Review · Reviewer_H9DP · 2024-05-10

**Rating:** 8
**Confidence:** 5
**Ethics Flag:** 1

**Summary:**

The paper presents an in-depth analysis into the performance of Large Language Models (LLMs), when asked to answer questions given irrelevant text as input (Irrelevant Context). This problem is crucial to ensuring the accuracy and faithfulness of LLMs, particularly what augmented with an Information Retrieval component, known as RALMs (Retrieval Augmented Language Models).

The paper builds upon previous works in cleverly defining 3 distinct types of irrelevant contexts (irrelevant, partially relevant and related). It also provides automatic generation methods (using prompted LLMs, Wikipedia and Knowledge Base) to generate such contexts automatically.

The experiments are rigorous, conducted on two QA benchmarks and using 4 LLMs of 3 different families (GPT, Llama, Gemini).
The paper is extremely well written and clearly presents and supports its scientific findings.

**Questions To Authors:**

Q1: Please add a brief discussion highlighting your differences and contributions from past work that also concern the robustness of LLMs to irrelevant contexts (Xie et al., 2023; Yoran et al., 2024). In particular, why isn’t the problem of “semantically related but irrelevant contexts” already captured in the top ranked Google passages, without ground truth answers, used by Yoran et al. [1]?

Q2: In Section 3.4 in the “Related Information” paragraph: the three variants of “Related Info” contexts are stated to be automatically generated by GPT-4 Turbo, based on triple of (subject, relation, object). There are two issues with this: (a) the prompts for generating the contexts are not provided in the paper or appendix and need to be included; (b) there is a concern the GPT-4 might create false contexts which can directly contradict the original questions grand truth answer. It is (very) briefly touched upon in A.3 but I strongly feel should be expanded and included in the main paper. The aspect of contradicting information should also be discussed in 3.4, with respect to the GPT-4 context generation.

Q3: In Section 3.5 it is stated that Facebook’s Contriver model [2] was used as the passage retriever. To the best of my knowledge, retrievers such as ColBERTv2 [3] are considered to be a better and more up-to-date retriever. Was there a particular reason for relying on Contriver instead of ColBERTv2 or Google Search API, as was done in prior works [1, 4]? I would appreciate including the reasoning behind using Contriver in the paper.

Q4: Table 2 shows that for GPT-3.5 and Gemini Pro, Unrelated Contexts drastically increase model uncertainty (UR), significantly more than the Partially Related Contexts (between 15-29 percentage points). This result is counter-intuitive, as Unrelated Contexts have less to do with the final answer than Partially Related ones. However, this UR result is not discussed anywhere in text. I would encourage the authors to include a brief discussion in the updated version.

Q5: In Section 4.4 “Limitations of Current Solutions” there is no mention of fine-tuning approaches to enhance robustness to irrelevant contexts [1]. This approach should also be addressed in this section, as it seems to have been successful at mitigating part of these issues, see [1]. To be clear, I do not feel that additional fine-tuning experiments are a must. I would appreciate adding a discussion on the role of fine-tuning in “LLM robustness to irrelevant contexts”.


[1] Ori Yoran, Tomer Wolfson, Ori Ram, and Jonathan Berant. Making retrieval-augmented language models robust to irrelevant context. ICLR, 2024.

[2] Unsupervised Dense Information Retrieval with Contrastive Learning (TMLR 2021)

[3] ColBERTv2: Effective and Efficient Retrieval via Lightweight Late Interaction

[4] Internet augmented language models through few-shot prompting for open-domain question answering

**Reasons To Accept:**

A1: The paper concerns an extremely important problem, crucial to the accuracy of LLMs in an information retrieval setting.

A2: The paper provides thorough experiments that clearly demonstrate its findings and contributions.

A3: The paper is extremely well written and clear.

**Reasons To Reject:**

R1: A few missing technical details and discussion points. Please see my “Questions to Authors” with respect to the issues that need to be corrected.

---

> ### Author Rebuttal · Authors · 2024-05-31
>
> Thanks for your efforts in reviewing!
>
> > Contributions
>
> Previous wisdom[1] uses top-ranked passages without ground truth answers or NLI model labeled as “irrelevant info”. However, such a process may still include relevant information due to model errors and hard to control of the information type. Moreover, for multi-hop questions, such passages might provide background for intermediate reasoning steps. By crafting fine-grained and controllable irrelevant information, we minimize the inclusion of relevant context and ensure a more systematic evaluation of robustness.
>
> > Generated information
>
> In our preliminary experiment, GPT4 may generate context contradicting ground truths. To address this, we add an instruction: “You are not allowed to claim or hint {subj'}’s {prop} is/isn’t {obj'}”. After manually sampling and examining the data, we have not observed this issue.
>
> We will provide the prompt in the next revision due to word limits.
> > Contriever
>
> In order to reuse the resources in PQA, we use Contriever they used. And Contriever is widely used in the community (~9M monthly downloads on HF).
> We further use ColBERTv2 to compute similarity scores between queries and our constructed information. It shows that our irrelevant information scores highly across different retrievers with high quality.
>
> ||Unrel.|PartRel.|Rel.|Wiki.|
> |-|-|-|-|-|
> |Scores|13.5|24.4|25.2|22.7|
> > Uncertainty Ratio
>
> LLMs might choose the uncertain option for security reasons within Unrelated Contexts. To illustrate this, we sample 100 data points from EQ to observe GPT3.5 Turbo's responses:
> ||Uncertain Count|
> |-|-|
> |Unrel.|73|
> |PartRel.|39|
> |Rel.|22|
>
> In uncertain situations within Unrelated Contexts, 57 out of 73 responses explain "The given information doesn't provide any details about the question", while the rest simply state the option. This indicates that GPT-3.5 identifies the unrelated contexts but expresses uncertainty for conservative reasons.
> > Fine-tuning(ft)
>
> Fine-tuning is a proven method to enhance LLMs’ robustness to irrelevant contexts[1]. In our paper, we focus on prompted-based methods to help discern irrelevant information.
> Furthermore, we sample 100 data points to test how llama2-13b and llama2-13b-peft-nq-retrobust (+ft)[1] perform with constructed related information. The MR indicates that the fine-tuned model shows improved robustness, but remains high, suggesting challenges still exist.
>
> ||MR|
> |-|-|
> |llama2 13b|61|
> |+ft|25|
>
> [1] https://arxiv.org/abs/2310.01558

---

> > ### Comment · Reviewer_H9DP · 2024-06-04
> > **Response to authors rebuttal**
> >
> > I thank the authors for their response. Based on their response, my score remains unchanged. I feel that the paper should be accepted, considering these changes will be included in the revised version.

---

> > > ### Author Response · Authors · 2024-06-06
> > >
> > > Thanks again for your engagement during the discussion. We are motivated by your encouraging feedback and recognition of our paper's contribution.

---

### Official Review · Reviewer_rBEp · 2024-05-11

**Rating:** 4
**Confidence:** 4
**Ethics Flag:** 1

**Summary:**

Authors design methods to construct semantically related but irrelevant passages to investigate LLM's robustness on single-hop factual QA problems. They draw several interesting conclusions:
1. LLM are more vulnerable to semantically related irrelevant passages compared with totally irrelevant passages.
2. more irrelevant passages, worse robustness
They also test LLM's robustness to irrelevant information under different settings such as QA task formats or prompting methods.

**Questions To Authors:**

please refer to the "reasons to reject".

**Reasons To Accept:**

Authors design methods to construct semantically related but irrelevant passages to investigate LLM's robustness on single-hop factual QA problems. They draw several interesting conclusions:
1. LLM are more vulnerable to semantically related irrelevant passages compared with totally irrelevant passages.
2. more irrelevant passages, worse robustness
They also test LLM's robustness to irrelevant information under different settings such as QA task formats or prompting methods.

**Reasons To Reject:**

1. This paper only considers the questions where the LLMs can correctly answer on their own and use the rate of answer modification as a metric. However, for RAG, what we really care about is the difficult questions, i.e., the ones that need retrieval. For these questions, we hope the LLM to be "distracted" by the "counter-memory" information and modify their original answers instead of "robust" to them. This work does not discuss them which makes the contributions limited.
2. experiments on 4.2 Quantity are not solid. More experiments, only 2 are given (3:0 vs 1:0, 3:1 vs 1:1), are needed to claim a tendency. Usually top-10 passages are concatenated to achieve the best retrieval-augmentation performance when using contriever as retriever. Current experiment only uses a maximum of 4 passages.
3. Free-form is the most common task setting in the LLM era and the most robust one regarding to table 4 (0.6% modification rate for gpt-3.5 turbo given relevant information), why still use multiple-choice for the main experiments?
4. methods used in "Limitations of Current Solutions" are naive. CoT and ICL are not designed to deal with irrelevant input and I believe there are many ad-hoc methods in RAG. I do not think it is ok to claimed that "current solutions for handling irrelevant information have limitations in improving the robustness of LLMs to such distractions."

---

> ### Author Rebuttal · Authors · 2024-05-31
>
> Thanks for your efforts in reviewing!
>
> > Difficult questions
>
> Our experiments consider both questions that LLMs answer correctly and incorrectly, thus covering "difficult questions".
> We haven't used the term "counter-memory" in our paper. It appears in reference[1], which analyzes its influence. Our focus is on the evaluation of "irrelevant" information.
>
> Unlike counter-memory, which suggests an exact answer different from memory-answer, irrelevant information cannot support a possible answer and it contains entities that distract LLMs. We aim to investigate whether LLMs are robust to these distractions.
>
> > Setting in 4.2
>
> As 10 passages is an ideal setting, acquiring large quantities of high-quality semantically related yet irrelevant information for specific questions is challenging. Designing a controllable experiment is also difficult due to identifying irrelevant entities.
> To address this, we utilize GPT4 to generate this information through triplets, ensuring quality and facilitating evaluation. To maintain diversity within our budget, we provide three variants of such information.
> Furthermore, in Sec4.4, we include 7 pieces of information to systematically evaluate LLMs' performance in a setting closer to real retrieval-augmentation.
>
> > Multiple-choice format
>
> The Multiple-Choice (M.C.) format simplifies answer evaluation by providing preset options, making it easier to assess LLMs' performance. In contrast, Free-Form (F.F.) responses are complex to parse, and aligning answers manually or using ChatGPT in main experiments is time-consuming and costly.
>
> M.C. format is also widely used in LLMs evaluation, e.g., MMLU, and CommonsenseQA.
>
> We manually sample 100 data points to evaluate GPT3.5's performance under F.F. format. The results are consistent with the main experiment using M.C. format: highly semantically related information is more likely to distract LLMs.
> ||Unrel.|PartRel.|Rel.|
> |-|-|-|-|
> |MR|1%|2%|6%|
> > Discussion about 4.4
>
> This paper focuses on LLMs' performance with irrelevant information. To our knowledge, few studies address improving LLMs’ robustness to irrelevant information through prompt engineering. Since ICL and CoT are fundamental in other RAG improvement methods[2,3], we chose these strategies for our evaluation due to their wide use. Please refer to the response to reviewer H9DP for the discussion about fine-tuning.
>
> [1] https://arxiv.org/abs/2305.13300
>
> [2] https://arxiv.org/abs/2402.18150
>
> [3] https://arxiv.org/abs/2402.13547

---

> > ### Author Response · Authors · 2024-06-06
> > **Kind Reminder from Authors**
> >
> > Hi Reviewer rBEp,
> >
> > Sincerely thanks again for your time and efforts in reviewing our submission.
> >
> > As the discussion period is ending soon, we wonder if there's any possibility that you can consider having a look at our reply. We would like to hear your feedback. If you have any other questions, please feel free to follow up anytime. We are committed to making any necessary revisions to further improve our work.
> >
> > Authors of Paper#148

---

### Decision · Program_Chairs · 2024-07-10

**Decision:**

Accept

**Comment:**

This paper investigates how retrieval-augmented generative models are affected by irrelevant context, and points out that these models are sensitive to "semantically relevant" irrelevant information.
Despite the fact that the experiments are done in a somewhat contrived setting, I agree with Reviewer H9DP that the conclusion illustrates an important issue regarding current language models: although they mostly rely on faithful knowledge bases in real practices, they are vulnerable to adversarial attacks.

In line with Reviewer jd2d, I also encourage the authors to take a step back and reconsider the term "semantically relevant irrelevant information."
Irrelevant information typically means information orthogonal to the task, and the proposal in this paper is closer to adversarial attacks, with a large amount of literature (e.g., Jia and Liang, 2017 and follow-up work) remaining undiscussed.

[comments from the PCs] Please take into account the AC and reviewer comments regarding positioning/context and terminology as you revise the paper. It will do a lot to improve the manuscript.

[At least one review was discounted during the decision process due to quality]